# Polyvinyl Butyral Addition Effects on Notched Charpy Impact Strength of Injection-Molded Glass Fiber-Reinforced Polypropylene

**DOI:** 10.3390/polym16243472

**Published:** 2024-12-12

**Authors:** Tetsuo Takayama, Yuuki Yuasa, Quan Jiang

**Affiliations:** Graduate School of Organic Materials Science, Yamagata University, Yamagata 990-8510, Japan; t231563m@st.yamagata-u.ac.jp (Y.Y.); t221291d@st.yamagata-u.ac.jp (Q.J.)

**Keywords:** additive, composites, glass fiber, injection molding, mechanical properties, sizing effect

## Abstract

Glass short fiber-reinforced thermoplastics (GSFRTPs) are a cost-effective alternative to other short fiber-reinforced thermoplastics (SFRTPs). Their excellent mechanical properties make them a suitable material for components that require rigidity and light weight in widely diverse fields, including transportation and office automation equipment. The melt-mixing process is used to shorten glass fibers. The notched impact strength of molded products is strongly affected by the fiber length. An important issue is how to conduct melt-molding processing while keeping the fibers long. In this regard, a survey of cases in which additives were used to increase the fiber length revealed no useful reports. However, a growing trend toward the reuse of plastic material wastes has emerged. When reusing GSFRTP wastes, the objective is to recycle the material as GSFRTPs. This promotion of the reuse of GSFRTPs necessitates the production of molded products with the fiber length maintained to the greatest extent feasible. Moreover, GSFRTPs should be recycled in a manner consistent with the original GSFRTPs. In recent years, there has also been a growing movement to reuse polyvinyl butyral (PVB) in accordance with Sustainable Development Goals (SDGs). It has been established that PVB can be extracted from the laminated glass state with high efficiency using mechanical methods. This study evaluated the mechanical properties of GSFRTPs with a PP matrix when PVB was added. The results show that the incorporation of PVB and maleic anhydride-modified PP in quantities of less than 1 wt% into GSFRTPs leads to sizing effects wherein the fibers are dispersed in bundles. Furthermore, this combination enhances the notched impact strength of the resulting molded product by 0.5 kJ/m^2^ at the maximum.

## 1. Introduction

Thermoplastics have found applications in a wide range of industries, from consumer goods to automotive manufacturing, because of their lightweight nature and superior moldability compared to metals and ceramics [1,2]. The lower melting temperatures of thermoplastics compared to those of metals and ceramics allow for melt-molding with low energy costs. Injection molding is a common method of thermoplastic molding because of its capacity to produce near-net-shaped components at a high rate [3,4].

Glass short fiber-reinforced thermoplastics (GSFRTPs) are a cost-effective alternative to other short fiber-reinforced thermoplastics (SFRTPs). GSFRTPs are frequently used in components that must have rigidity and lightweight design for applications in a multitude of industries, including transportation and office automation [5]. GSFRTPs can be achieved through a process of melting and mixing glass fibers as short fibers with thermoplastics. In melt mixing, a shear load is applied during compression to disperse the filler material efficiently and uniformly. Reportedly, this compressive and shear loading causes a pressure loss in the glass fibers [6,7,8,9]. In other words, glass fibers are shortened by melt mixing. In fields where GSFRTPs are utilized, impact resistance is frequently a requisite quality, in addition to stiffness. The notched Charpy impact test is often employed as an indicator of this quality [10]. Because the fiber length strongly affects the notched impact strength of molded products, an important issue is how to conduct the melt-mixing process while keeping the fibers long. Investigations into the lengthening of residual fibers have been conducted, mainly from the viewpoint of molding processes [11,12,13]. For example, studies have examined the optimization of the molding process conditions during melt mixing [11,12], the screw configuration [12], and the feeding method [13]. From the perspective of the materials themselves, the correlation with the viscosity of the base material has also been studied. After Durin et al. proposed a buckling parameter that reflects the fiber orientation angle, viscosity, and shear rate, it was used to predict the fiber pressure loss which occurs in twin-screw extrusion [13]. A survey of reports examining the use of additives to increase the fiber length revealed no useful findings.

Conversely, the authors highlight that if the fiber is manufactured to an excessive length, the impact resistance may be adversely affected [14]. This is due to the fact that when fibers exceed the critical fiber length, the fiber pullout length, which contributes to the strength and notched impact strength, is dependent on the critical fiber length. In such instances, the enhancement of these properties can be anticipated by increasing the fiber diameter or weakening the interfacial shear strength.

There is a growing trend toward the reuse of plastic material wastes, with SFRTPs being reused to an increasing degree by separating and extracting resin and fiber from its wastes [15,16,17]. However, glass fibers are less expensive than other fibers. Therefore, the previously described pressure loss of fibers during melt processing renders the trend of separating and reusing glass fibers from GSFRTP waste commercially unfeasible. Therefore, when reusing GSFRTP waste, it is desirable to recycle the material as a GSFRTP. However, if the material is recycled as a GSFRTP, then it must be melt-molded again, which further shortens the fiber length [6]. This results in a reduction in the static strength and notched impact strength of the molded product. The notched impact strength is notably influenced by the fiber length [18]. Consequently, to promote the reuse of GSFRTPs, it is necessary to maintain the fiber length as long as possible in the molded product.

Polyvinyl butyral (PVB) is frequently used as an adhesive for laminated glass in automobiles [19]. Upon the scrapping of these vehicles, the PVB is incinerated, resulting in the generation of an incineration residue that is subsequently disposed of in landfills. Landfill disposal represents an important environmental concern because of its potential for pollution. Therefore, minimizing its occurrence is imperative.

In recent years, there has been a movement to reuse this material in the context of Sustainable Development Goals (SDGs) [17,20]. It has been demonstrated that PVB can be extracted from its laminated glass state with high efficiency using mechanical methods [21,22]. Consequently, it is not difficult to envision that PVB extracted from waste automobiles could be a target for material recycling. For example, Cervantes et al. reported that surface friction properties can be improved by compounding PVB from waste automobiles with glass fiber-reinforced polyamide [23].

Furthermore, PVB reportedly reacts compatibly when blended with polypropylene (PP) by adding maleic anhydride-modified PP (MAH-PP) [24]. It is therefore anticipated that the use of PVB and MAH-PP together as an additive in glass fiber-reinforced thermoplastic polypropylene will strengthen the fiber–matrix interface and improve strength and impact resistance.

For this study, the notched Charpy impact strength of GSFRTPs with a PP matrix was evaluated with the addition of PVB. The results demonstrated that the combination of PVB and maleic anhydride-modified PP caused enhanced fiber lengthening and a notable increase in the notched impact strength of the molded product.

## 2. Materials and Methods

### 2.1. Materials

Polypropylene (PP, Novatec MA1B; Japan Polypropylene Corp., Tokyo, Japan) was used as the matrix. The melt volume rate (MVR), which serves as a measure of the viscosity of polypropylene (PP), has been determined to be 27 cm^3^/10 min at 230 °C and 2.160 kgf. Glass fiber (GF, ECS 03 T351; Nippon Electric Glass Co., Ltd., Otsu, Japan) was used as the fiber. The fiber diameter is 13 μm. Its length before the application of melt mixing was 3 mm. Recycled PVB (R-PVB) extracted from the side glass of end-of-life vehicles and maleic anhydride-modified polypropylene (MAH-PP, SCONA TSPP10213; BYK Additives & Instruments Co., Ltd., Wesel, Germany) were used as additives. The MVR of R-PVB is 11 cm^3^/10 min at 230 °C and 2.160 kgf. The MVR of MAH-PP is 40 cm^3^/10 min at 170 °C and 2.160 kgf.

### 2.2. Melt-Mixing Process

The previously described materials were introduced into a twin-screw melt extruder (IMC0-00; Imoto Machinery Co., Ltd., Kyoto, Japan). The resulting strands were pelletized using a pelletizer (cold-cut pelletizer; Toyo Seiki Co., Ltd., Tokyo, Japan) to obtain 3 mm long composite pellets. The melt temperature was set at 230 °C. The screw speed was maintained at 60 rpm. The fiber content was adjusted to 30 wt%, the amount of PVB added varied from 0, 0.5, 1, and 5 wt%, and the amount of MAH-PP added was fixed at 5 wt%.

### 2.3. Injection Molding

Figure 1 depicts an overview of the injection molding machine used. The composite pellets were subsequently filled into an ultra-compact electric injection molding machine (C, Mobile0813; Shinko Sellbic Co., Ltd., Tokyo, Japan) for injection molding. The injection molding conditions are presented in Table 1. The injection temperature and mold temperature were fixed, respectively, at 230 °C and 50 °C. The holding pressure was set to the value at which optimal products were obtained. Additionally, molded products were produced by diluting PP/GF pellets produced through extrusion molding during injection molding, with pellets produced with the composition described previously. The matrix phase was modified to adjust the GF content to 10 and 20 wt%. The molded product was a beam 50 mm long, 2 mm thick, and 5 mm wide. During molding, the flow rate was also adjusted to ensure that the resin meeting point (welding zone) was centered within the specimen.

### 2.4. Short Beam Shear Tests

Three-point bending tests were conducted on a compact universal mechanical testing machine (MCT-2150; A&D Co., Ltd., Tokyo, Japan) using a beam-shaped molded product with welds and a distance of 10 mm between fulcrum points. The loading speed was set at 10 mm/min. The specimens were positioned according to the orientation shown in the schematic representation of the test (Figure 2). The load–deflection curve was differentiated by deflection to obtain stiffness, and from the load, the average shear stress τ was determined by the following Equation (1) to produce a stiffness-averaged shear stress curve [25].

(1)
$$\tau =\frac{3P}{4A}$$

where P is the load, and A is the cross-section area. The interfacial shear strength (IFSS) was determined based on the method proposed by Quan et al. [25], using the stiffness-averaged shear stress curve obtained.

### 2.5. Tensile Tests

Uniaxial tensile tests were conducted using a small universal mechanical testing machine (FSA-1KE-1000N-L; Imada Co., Ltd., Toyohashi, Japan) in accordance with ISO 527-1 using a beam-shaped molded product with a welding zone [26]. A schematic representation of the test is provided in Figure 3. The distance between the chucks was 22.5 mm, and the loading speed was 1.5 mm/min. A nominal stress–true strain curve was obtained from the load–displacement curve. The maximum nominal stress obtained from this curve was determined to be the weld strength.

### 2.6. Notched Charpy Impact Tests

A notch was introduced in the center of the specimen by the machining of a beam-shaped molding without a welding zone. Then a Charpy impact test was performed using a Charpy impact testing machine (Mys-shikenki Co., Ltd., Osaka, Japan) in accordance with ISO 179-1 [27]. A schematic representation of the test is provided in Figure 4. The loading speed was 2.91 m/s. The spun length was 40 mm. The Charpy impact strength of the specimens was found by calculating the absorbed energy U from the obtained swing angle based on the following Equation (2) [27].

(2)
$$a_{iN}=\frac{U}{B\left(W-a\right)}$$
 In this equation, B stands for the molded product thickness. In addition, W denotes the width of the molded product and a represents the notch depth.

### 2.7. Fracture Surface Observation

The fracture surface obtained from the Charpy impact test was observed using a scanning electron microscope (Tiny-SEM 510; Technex Lab Co., Ltd., Tokyo, Japan) to identify the fracture morphology and the state of dispersion of the PVB.

### 2.8. Residual Fiber Length Evaluation

An X-ray CT system (ScanXmate-D225RSS270; Comscantecno Co., Ltd., Yokohama, Japan) was employed to examine the skin layer area of a molded product devoid of welds, obtained through injection molding. Using the outcomes of the observed skin layer, more than 1000 fibers were extracted through image analysis. Then, the length of each was determined. The mean fiber length was evaluated as the characteristic L_f_ value. This L_f_ value was used to assess the mechanism of Charpy impact strength development.

### 2.9. Fiber Orientation Observation

An X-ray CT system (ScanXmate-D225RSS270; Comscantecno Co., Ltd., Yokohama, Japan) was used to observe the fiber orientation in the core layer region of an injection-molded product. The observed results were then used to extract over 1000 fiber orientation angles with respect to the width direction of the molded product, with the average value evaluated as the characteristic value.

## 3. Results

### 3.1. GF and PVB Content Dependences on Interfacial Mechanical Properties of PP/GF

Figure 5 shows the dependence of the IFSS of PP/GF/PVB and PP/MAH-PP/GF/PVB on the GF and PVB content. The characteristic values used for the figure are presented in the IFSS column of Table 2. The interfacial shear strength with and without MAH-PP tended to decrease concomitantly with increases in the GF content; the addition of MAH-PP increased the interfacial shear strength. The decrease in interfacial shear strength with increases in the GF content slowed. These phenomena, which are elaborated upon in Section 4.1, are postulated as attributable to the reduction in the interfacial interaction force, concomitant with an increase in fiber content.

The interfacial shear strength tended to increase with the addition of 0.5 wt% PVB to PP/GF. The value decreased with the addition of 5 wt% PVB. By contrast, the interfacial shear strength decreased with the addition of 0.5 wt% PVB to PP/GF/MAH-PP. No marked change was observed with increases in the PVB content.

Figure 6 presents the dependence of PP/GF/PVB and PP/MAH-PP/GF/PVB on the GF and PVB content on the weld strength. The characteristic values used for the figure are from the σ_w_ column of Table 2. The weld strength with and without MAH-PP tended to decrease concomitantly with increases in the GF content. Also, the addition of MAH-PP tended to increase the weld strength. The decreasing trend in weld strength with increases in the GF content slowed. This trend is believed to result from the same mechanism as the IFSS trend, as discussed further in Section 4.1. This trend is attributable to the fact that the interfacial interaction force decreases concomitantly with increases in the fiber content.

The addition of PVB to PP/GF tended to decrease the weld strength, which became more pronounced as the amount of PVB increased. By contrast, the addition of PVB to PP/GF/MAH-PP exhibited a tendency to decrease the weld strength, which was not affected strongly by increasing the amount of PVB added.

The results indicate that the addition of MAH-PP to PP/GF strengthens the interface, irrespective of the GF content. Furthermore, the addition of PVB alone to PP/GF at 0.5 wt% strengthens the interface, whereas the addition of 1 wt% or more weakens the interface. These findings suggest that the weakening of the interface with increases in the PVB content can be suppressed using MAH-PP in combination with PP/GF.

### 3.2. PVB Effects on the Notched Charpy Impact Strength

Figure 7 portrays the results of the Charpy impact strength evaluation. The characteristic values used for the figure are presented in the a_iN_ column of Table 2. The Charpy impact strength tended to increase concomitantly with increases in the GF content. The details of this phenomenon are discussed further in Section 4.3, but this phenomenon is often observed when the impact energy is dissipated through fiber pullout. The addition of MAH-PP tended to increase the Charpy impact strength.

When PVB was added to PP/GF, the value decreased at 0.5 and 1 wt% and increased at 5 wt%. When PVB was added to PP/GF/MAH-PP, the value increased at 0.5 wt% and decreased at 1 and 5 wt%. The value tended to decrease at 1 and 5 wt%.

These results indicate that adding MAH-PP to PP/GF increases the Charpy impact strength. Furthermore, when only PVB is added to PP/GF, the same value increases when the amount of PVB added is high. A synergistic effect with PVB can be expected when the PVB content is low and when MAH-PP is used in combination with PP/GF.

## 4. Discussion

### 4.1. Relations Between Interfacial Shear Strength and Weld Strength

A comparison of Figure 1 and Figure 2 reveals similar trends for the weld strength and interfacial shear strength. Figure 8 depicts the correlation between the weld strength and interfacial shear strength. As this figure shows, a proportional relation exists between the weld strength and interfacial shear strength, with a slope of approximately 3.

The interfacial shear strength is defined as the shear stress at the interface when a shear deformation begins to occur at the interface. The interfacial strength is defined as the stress at which the interface delaminates because of the normal stress at the interface. The shear stress applied in the short beam shear test is regarded as an equi-triaxial shear stress because of its conjugate nature [28]. After substituting the principal stress conditions of the equi-triaxial shear stress state into the von Mises or Tresca yield conditions and rearranging them, the following relation between the interfacial strength σ_i_ and interfacial shear strength τ_i_ is shown, as in Equation (3).

(3)
$$\sigma_{i}=3\tau_{i}$$
 The weld strength is the tensile strength obtained when the fiber is oriented as perpendicular to the loading direction. The weld strength is synonymous with the interfacial strength described above because yielding at this time results from interfacial debonding [29]. From the verification presented above, it is proven theoretically that the relation in Equation (3) holds between the weld strength and the interfacial shear strength.

Takayama also reported that the interfacial strength is obtainable from the following Equation (4) [29].

(4)
$$\sigma_{i}=\left(\alpha_{m}V_{m}-\sum_{k=1}^{n-1}\alpha_{k}V_{k}\right)∆TE_{cT}+\gamma_{i}S_{f}$$
 Therein, α is the average linear expansion coefficient, ΔT represents the difference between the injection molding temperature and the test temperature, V stands for the volume content, γ_i_ denotes the interfacial interaction force, and S_f_ expresses the specific surface area at the fiber–matrix interface. Also, n is the number of compositions and the subscript m denotes the matrix. E_cT_, which is the elastic modulus of the composite when the fibers are oriented as perpendicular to the loading direction, is obtained using the following Equation (5) [29].

(5)
$$E_{cT}={\left(\sum_{j=1}^{n}\frac{V_{j}}{E_{j}}\right)}^{-1}$$
 Taken together, these equations and results suggest that the decrease in weld strength and interfacial shear strength with increases in the GF content is mainly attributable to the decrease in the interfacial interaction force. This is probably because of the fact that, as the GF content increases, the distance between fibers becomes closer, resulting in a stronger interaction force between fibers and consequently a lower interfacial interaction force. For example, if the interaction force occurring in a single fiber is expressed as γ_0_ and the interaction force occurring between fibers is expressed as γ_f_, then it can be expressed in terms of the following Equation (6).

(6)
$$\gamma_{i}=\gamma_{0}+\gamma_{f}$$
 In this equation, γ_f_ is regarded as negative from the perspective of the subject. To express that the inter-fiber distance increases in the negative direction as it becomes closer, γ_f_ must be a function of the inverse of the inter-fiber distance. In other words, Equation (6) can be rewritten as the following Equation (7).

(7)
$$\gamma_{i}=\gamma_{0}+f\left({\langle L_{f}\rangle }^{-1}\right)$$
 Therein, L_f_ is the inter-fiber distance. Interaction forces that occur even where fibers are separated by micrometers include capillary and intermolecular forces. Capillary forces are proportional to the inverse of the distance between fibers, whereas intermolecular forces are proportional to the inverse square of the distance between fibers. Based on the points raised above, γ_i_ can be expressed by the following Equation (8).

(8)
$$\gamma_{i}=\gamma_{0}+\gamma_{c}{\langle L_{f}\rangle }^{-1}+\gamma_{im}{\langle L_{f}\rangle }^{-2}$$
 Here, γ_c_ and γ_im_ are coefficients. Based on the discussion above, the increase in the interfacial shear strength and weld strength because of the addition of MAH-PP can be attributed to the increase in γ_0_ and to the suppression of the decrease in γ_f_, which results from a fiber–fiber interaction.

### 4.2. PVB Addition Effects on the Interfacial Shear Strength and Weld Strength of PP/GF

The addition of PVB to PP/GF caused a decrease in the weld strength and interfacial shear strength. Moreover, this trend was more pronounced as the amount of PVB added increased, probably partly because of the fact that the coefficient of the linear expansion of PVB is larger than that of PP and GF, resulting in a smaller thermal residual strain term in Equation (4). To clarify this consideration, the flexural modulus of PP and PVB was determined using a three-point bending test. Furthermore, the average linear expansion coefficient was calculated based on Baker’s experimental formula in Equation (9) [30].

(9)
$$\alpha =\sqrt{\frac{15}{E_{11}(MPa)}\times 10^{-6}}$$
 Here, E_11_ is the apparent modulus of elasticity when subjected to pure tensile deformation, which in this study, is the flexural modulus [31]. The average linear expansion coefficients for PP, PVB, and GF are presented in Table 3. For the PP, beam specimens of injection-molded PP were used. For the PVB, test pieces were made by layering multiple layers of film produced using a heat press machine and by then forming them into beams. The table shows that the average linear expansion coefficient of PVB is more than 80 times larger than that of PP, which is believed to be the reason why the thermal residual strain term shown in Equation (4) is smaller even when the PVB content is small.

Furthermore, the interfacial interaction force is quantified and discussed using Equation (4). Figure 9 shows the dependence of the interfacial interaction force γ_i_ shown in Equation (4) on the GF content and PVB content. The characteristic values used for the figure are presented in the γ_i_ column of Table 2. As this figure shows, γ_i_ decreases with the addition of 0.5 wt% PVB, and increases with the addition of 1 wt% or more PVB. This trend was obtained irrespective of the GF content or MAH-PP.

Figure 10 shows examples of the X-ray CT imaging results of the core layer when the GF content is 20 wt%. When 0.5 wt% or 1 wt% PVB was added to PP/GF, the fibers were dispersed in a slightly uneven manner, which suggests that the fibers are localized together. In fact, the localization of GF might have caused a tendency for γ_i_ to decrease because of the closer fiber-to-fiber distance. By contrast, when the quantity of PVB incorporated was augmented to 5 wt%, the fiber dispersion exhibited a comparable configuration to that of PP/GF, accompanied by an inclination for γ_i_ to augment. Intermolecular forces include ion–ion interactions, hydrogen bonds, dipole interactions, and van der Waals forces. Reportedly, PVB possesses high adhesion properties because of hydrogen bonding with GF as a result of its large number of hydroxyl groups [21]. In other words, the higher proportion of PVB in the mixture would have augmented the intermolecular forces between PVB and GF, which would have caused an increase in γ_i_. In consideration of the evidence presented earlier, the observed decline in the interfacial shear strength and weld strength with an increasing PVB addition is postulated to be the consequence of a complex interplay among the propensity of γ_i_ to undergo alteration because of fiber localization, the intensification of intermolecular forces resulting from PVB addition, and the diminution in the thermal residual strain at the interface attributable to the substantial linear expansion coefficient of PVB.

### 4.3. PVB Addition Effects on the Notched Charpy Impact Strength of PP/GF

To provide a quantitative explanation for the change in the notched impact strength resulting from the addition of PVB, it is first necessary to confirm the fracture morphology produced in the notched impact test. Figure 11 depicts a specimen following a notched Charpy impact test. The morphology depicted in this figure was observed in all the compositions examined for this study. This figure shows that the specimen undergoes cracking from the tip of the notch, with the crack propagating in the width direction of the specimen, ultimately leading to fracture. The fracture surface displays numerous fibers, indicating that the injection-molded specimens examined for this study were subject to fiber pullout caused by impact loading, which dissipated the impact energy.

Quan et al. reported that when the fiber orientation angle is not perpendicular to the loading direction, the Charpy impact strength of GSFRTPs is explainable by the following Equation (10) [18].

(10)
$$a_{iN}=\tau_{i}\cos\varphi S_{f}l_{p}^{2}$$
 Therein, φ represents the average orientation angle of the fibers in the core layer region. Also, l_p_ denotes the fiber pullout length, which is determined approximately by the following Equation (11), based on the critical fiber length L_c_ [14].

(11)
$$l_{p}=\left\{\begin{array}{c} \frac{L_{F}}{2}\;\mathrm{for}\;L_{F}<L_{c}\\ \frac{L_{c}}{2}\;\mathrm{for}\;L_{F}>L_{c} \end{array}\right.$$
 In this equation, L_F_ denotes the remaining fiber length. Also, L_c_ stands for the critical fiber length, which is obtained using Equation (12) [32] as follows:
(12)
$$L_{c}=\frac{d\sigma_{FB}}{2\tau_{i}}$$

where d stands for the fiber diameter and σ_FB_ expresses the fiber tensile strength. Equations (10)–(12) can be rearranged into Equation (13) as follows:
(13)
$$a_{iN}=\left\{\begin{array}{c} \frac{\tau_{i}V_{f}L_{F}^{2}}{d}\cos\varphi \mathrm{for}\;L_{F}<L_{c}\\ \frac{dV_{f}\sigma_{FB}^{2}}{4\tau_{i}}\cos\varphi \mathrm{for}\;L_{F}>L_{c} \end{array}\right.$$

where S_f_ in Equation (10) is the specific surface area of the interface acting on fiber pullout, which is obtained using the following Equation (14) [14].

(14)
$$S_{f}=\frac{4V_{f}}{d}$$
 From Equation (13), it is apparent that the following Equation (15) is correlated.

(15)
$$a_{iN}\propto \left\{\begin{array}{c} \frac{\tau_{i}}{d}\mathrm{for}\;L_{F}<L_{c}\\ \frac{d}{\tau_{i}}\mathrm{for}\;L_{F}>L_{c} \end{array}\right.$$

Figure 12 depicts a graph of a_iN_ on the vertical axis and τ_i_ on the horizontal axis according to Equation (15). These figures show a negative correlation between a_iN_ and τ_i_ with and without the MAH-PP or PVB addition. A slight variation among the compositions indicates that a_iN_ is influenced not only by τ_i_ but also by the specific surface area and fiber orientation angle at the interface. In light of Equation (15), it can be posited that the residual fiber lengths of the compositions considered in this study are all longer than the critical fiber length. To test this hypothesis, we employed Equation (10) to back-calculate l_p_ using a_iN_ obtained from the Charpy impact test and φ obtained from X-ray CT observations. Then, we examined the relation between this result and the average fiber length obtained from X-ray CT observations. Figure 13 shows the average fiber orientation angles obtained using an image analysis applied to the X-ray CT imaging results. The characteristic values used for the figure are presented in the φ column of Table 2. Here, the orientation angle is obtained based on the width direction of the molded product. Without MAH-PP, the fiber orientation angle showed a slight increase with the addition of PVB to compositions with a low GF content (10 wt%). No clear change in the fiber orientation angle was observed in other compositions. When MAH-PP was added, the fiber orientation angle did not appear to change markedly with the addition of PVB. These results indicate that the addition of only PVB to PP/GF 10 wt% makes the fibers slightly more oriented parallel to the flow direction. The observed trends indicate that the filling and cooling solidification behavior resulting from injection molding is not markedly influenced by the GF content or the presence of additives. One potential explanation for this finding is that the injection molding speeds used for this study fell within a range where shear thinning was considerable, and the composition had only a minimal effect on the results.

Figure 14 portrays the correlation between the average fiber length L_f_ of GF in the skin layer region and the concentration of GF and PVB. The characteristic values used for the figure are presented in the L_f_ column of Table 2. The fiber length remained consistent throughout the composition range investigated in this study, exhibiting minimal variation. The reason for this slight variation might be related to the molding process performed in this paper. As described in this paper, the GF content in the injection-molded product is controlled by diluting composite material pellets, which have a constant GF content, with base material pellets during injection molding, which indicates that the fiber length of GF present in the composite pellets obtained by melt mixing is unchanging. Consequently, the alteration in the GF fiber length within the injection-molded product is attributable to the shear stress received during injection molding. Given that the injection molding speeds employed in this study fall within the shear strain rate range where shear thinning occurs to a marked degree, as stated previously, the resin viscosity at the time of mold inflow is not influenced strongly by the composition. Consequently, it can be posited that the composition only slightly affects the fiber length of GF in the injection-molded product. Figure 15 shows the relation between l_p_ and L_f_. When L_f_ is shorter than L_c_, the relation follows the dotted line in Figure 15 from Equation (11). The results obtained from this study deviate from the relation shown by the dotted line in Figure 15 for all compositions. This finding shows that the residual fiber lengths of the compositions considered in this paper are all longer than the critical fiber length.

If the residual fiber length is greater than the critical fiber length, then σ_FB_ is obtainable using Equation (13). σ_FB_ for each composition is shown in Figure 16. The characteristic values used for the figure are presented in the σ_FB_ column of Table 2. The σ_FB_ tended to decrease concomitantly with increases in the GF content. This trend became weaker with the addition of MAH-PP. The strength of the glass fiber presented in Figure 16 is 280–500 MPa, which is markedly lower than the reported tensile strength of 1500 MPa for glass fiber [33]. This finding indicates that other factors are necessary in addition to the items considered in Equation (11) to obtain the tensile strength of the fiber using Equation (13). As described herein, we specifically examine the fiber orientation angle. Through trial and error, we found that the fiber strength can be explained approximately using the following Equation (16).

(16)
$$\sigma_{FB0}=\frac{4\sigma_{FB}}{\sin^{2}\varphi }$$


Figure 17 shows the σ_FB0_ strength of the fiber obtained using Equation (16). The characteristic values used for the figure are presented in the σ_FB0_ column of Table 2. The results show that σ_FB0_ varied with the GF and PVB content and MAH-PP addition, but the values remained in the range of 1500–2100 MPa, which was similar to results reported earlier in the relevant literature. Therefore, a correction by Equation (16) was judged to be reasonable [33]. The discrepancies between the compositions shown in the figure might be attributed to alterations in surface mechanical properties resulting from chemical reactions triggered by the incorporation of PVB and MAH-PP, in addition to modifications in interfacial tension caused by shifts in interaction forces. These results indicate that the glass fibers in the samples examined for this study are sufficiently longer than the critical fiber length.

Figure 18 presents the results of the fracture surface observations after Charpy impact testing was performed on a composition with a GF content of 20 wt%. It was observed that the addition of 0.5 wt% PVB produced fibers with PVB adhering to the surface of the fibers. This observation, together with the results presented in Figure 10, suggests that the scattering of PVB-attached GFs is a factor that brings the distance between fibers closer. This phenomenon might have caused the decrease in γ_i_ shown in Section 4.2. In addition, when considered together with MAH-PP, GF was observed to be dispersed in a bundle state. This dispersion is probably attributable to the higher interfacial interaction force caused by the addition of MAH-PP. The addition of 0.5 wt% of PVB is thought to have brought the distance between fibers closer. Also, the addition of MAH-PP increased the interfacial interaction force, causing GF to bundle and suppressing fiber disassembly caused by the shear stress received during melt mixing. These effects demonstrate that d in Equation (13) has increased. From Equation (15), when L_f_ > L_c_, a_iN_ is regarded as having increased because of the proportional relation between a_iN_ and d. A similar dispersion state was observed for a PVB content of 1 wt%, but a slightly smaller φ was obtained. Therefore, a_iN_ was not regarded as having improved.

However, when the amount of PVB increased to 5 wt%, Figure 10 shows no evidence of GF bundling. Moreover, Figure 18 presents an increase in the amount of GF with PVB attached. These outcomes suggest that the increase in the amount of PVB adhered to the GFs caused an increase in the number of GFs to which the PVB adhered, thereby homogenizing the fiber-to-fiber interaction forces and suppressing the uneven distribution of GFs. Furthermore, when the amount of PVB added reached 5 wt%, it was observed not only to adhere to the surface of GFs, but also to be dispersed in a phase-separated manner in the matrix. This phase structure was no longer observed with the addition of MAH-PP, which suggests that MAH-PP acts as a reaction compatibilizer between PP and PVB. One possible reaction found in this study is the esterification reaction of the acid anhydride functional group of maleic anhydrides with the hydroxyl group of PVB. The reaction products generated by this process localize at the interface between PP and PVB, resulting in a decrease in the interfacial tension and micro dispersion of the PVB phase. Furthermore, the entanglement of the reaction products with the molecular chains of each phase creates new intermolecular friction. This friction mechanism is regarded as being established in the melting state. Therefore, the excess addition of PVB in combination with MAH-PP is assumed to increase the melt viscosity of the matrix. Figure 9 shows that the interfacial interaction force is highest when the amount of PVB added is 5 wt%. Also, the melt viscosity is assumed to be higher than when the amount of PVB added is 1 wt% or less. The addition of MAH-PP might increase the melt viscosity further, which might then strongly promote the shortening of the fibers produced during melt mixing. These factors might have caused the shortest fiber length at 5 wt% PVB.

These results and discussion indicate that the addition of a small amount of PVB acts as a GF sizing agent, whereas excessive addition promotes fiber shortening. The effect of a small amount of PVB on the residual fiber length of GF was more pronounced when used in combination with MAH-PP. This finding suggests that a certain degree of higher interfacial interaction force makes the fibers more susceptible to shear conditions received during melt mixing in a bundled state. Consequently, fiber disassembly is more likely to be suppressed. In other words, controlling the interfacial interaction force without changing the amount of PVB added is an important means of improving the PP/GF Charpy impact strength effectively by adding PVB.

The results obtained from this study demonstrate that the addition of a small amount of PVB extracted from automobile waste can improve the Charpy impact strength because of the ubiquity of GF. This effect is thought to depend on the ratio of GF to PVB, but the amount of added PVB that was effective in the study described in this paper was less than 1 wt%. The optimal amount of PVB to be added is expected to be different for compositions with a higher GF content. In future studies, we would like to verify the optimal PVB addition amount for compositions with an even higher GF content. We would also like to verify whether the effect of adding PVB obtained this time is obtainable even when PP with different viscosities is used as the base material.

## 5. Conclusions

This study investigated the effects on changes in the notched Charpy impact strength obtained by adding polyvinyl butyral (PVB) extracted from automobile waste to short glass fiber-reinforced polypropylene (PP/GF). The findings suggest that, under certain conditions, the notched Charpy impact strength might be improved. The addition of small amounts of PVB alone had the effect of strengthening the PP/GF interface, but it also contributed to the uneven distribution of glass fibers. Furthermore, findings indicate that the combination of MAH-PP with the notched impact strength can be enhanced by up to 0.5 kJ/m^2^ while maintaining the fibers’ bundle-like dispersion. However, it was also established that the excessive addition of PVB leads to fiber length shortening and to a suppressed improvement in te notched impact strength.

## Figures and Tables

**Figure 1 polymers-16-03472-f001:**
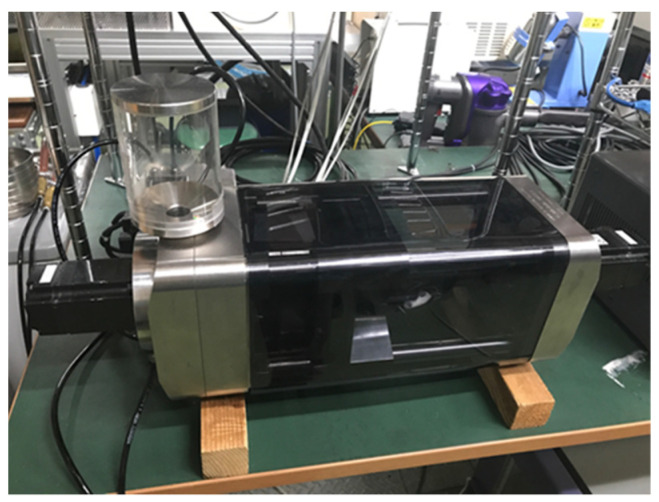
Injection molding machine used for this study.

**Figure 2 polymers-16-03472-f002:**
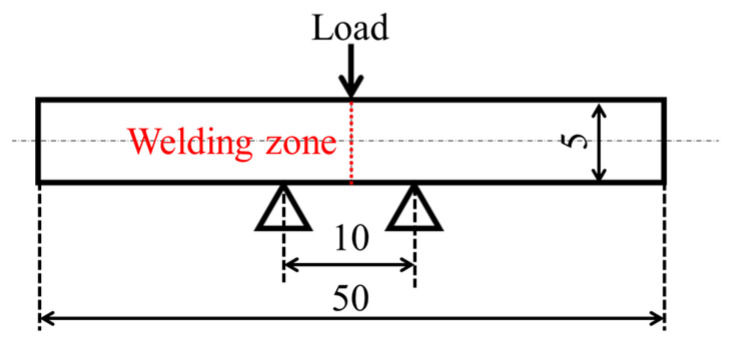
Schematic representation of the short beam shear test (Unit: mm).

**Figure 3 polymers-16-03472-f003:**
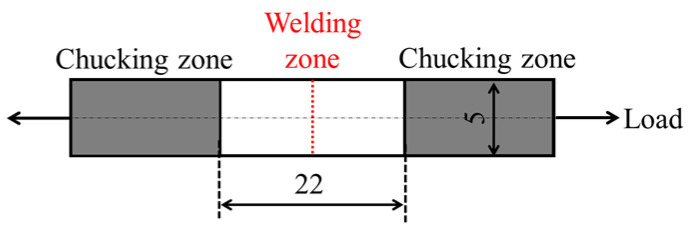
Schematic representation of the tensile test (Unit: mm).

**Figure 4 polymers-16-03472-f004:**
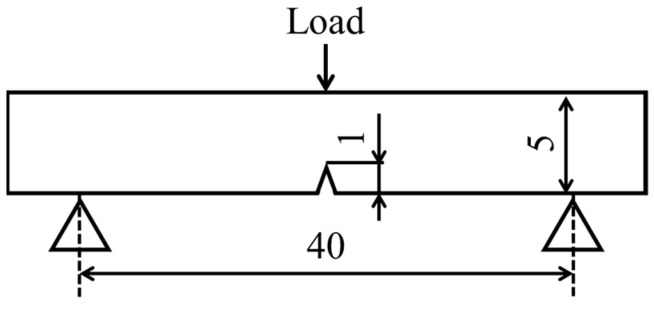
A schematic representation of the tensile test (Unit: mm).

**Figure 5 polymers-16-03472-f005:**
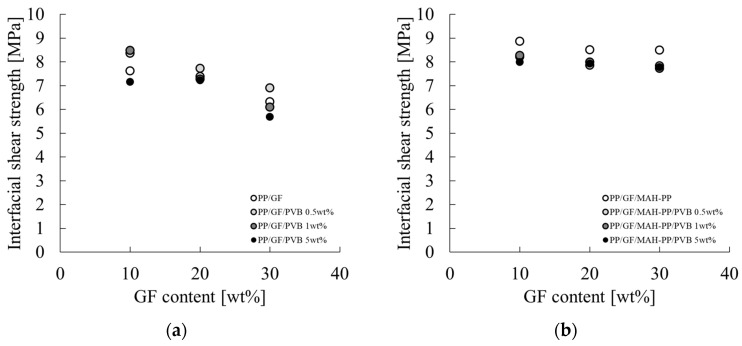
GF content dependences on the interfacial shear strength of (**a**) PP/GF/PVB and (**b**) PP/GF/MAH-PP/PVB composites.

**Figure 6 polymers-16-03472-f006:**
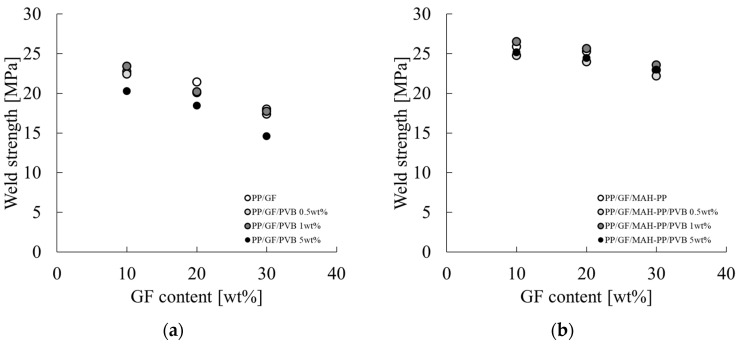
GF content dependences on the weld strength of (**a**) PP/GF/PVB and (**b**) PP/GF/MAH-PP/PVB composites.

**Figure 7 polymers-16-03472-f007:**
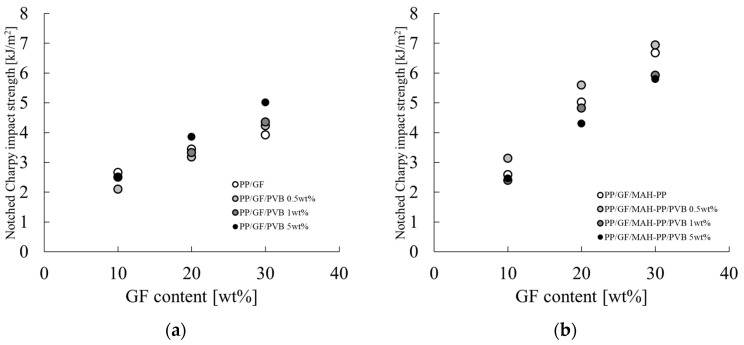
GF content dependences on the notched Charpy impact strength of (**a**) PP/GF/PVB and (**b**) PP/GF/MAH-PP/PVB composites.

**Figure 8 polymers-16-03472-f008:**
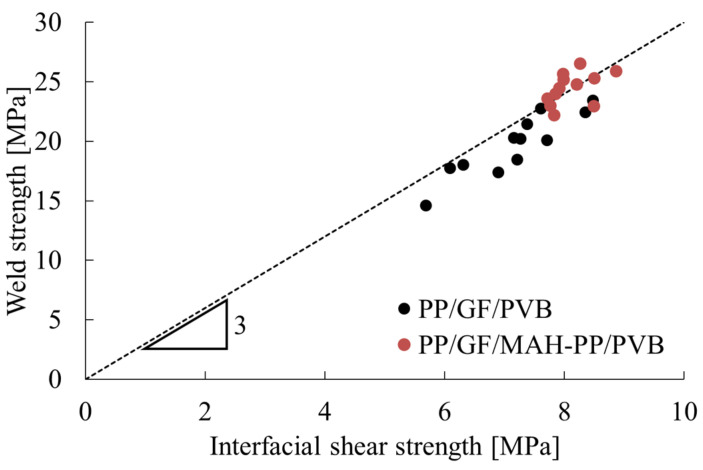
Relation between weld strength and interfacial shear strength. The triangles in the figure represent the slope of the dotted line, and the numbers represent the magnitude of the slope.

**Figure 9 polymers-16-03472-f009:**
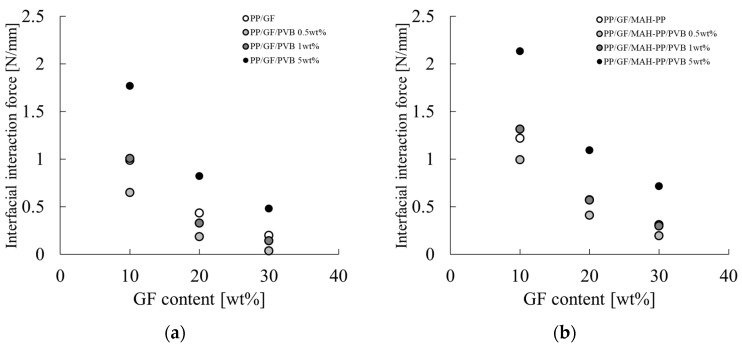
GF content dependences on the interfacial interaction force of (**a**) PP/GF/PVB and (**b**) PP/GF/MAH-PP/PVB composites.

**Figure 10 polymers-16-03472-f010:**
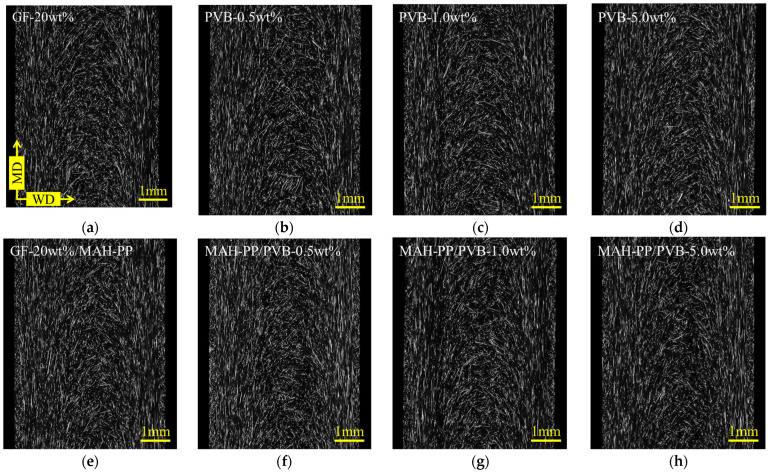
X-ray CT imaging results of the core layer obtained when GF content is 20 wt%. (**a**) PP/GF, (**b**) PP/GF/PVB 0.5 wt%, (**c**) PP/GF/PVB 1 wt%, (**d**) PP/GF/PVB 5 wt%, (**e**) PP/GF/MAH-PP, (**f**) PP/GF/MAH-PP/PVB 0.5 wt%, (**g**) PP/GF/MAH-PP/PVB 1 wt%, and (**h**) PP/GF/MAH-PP/PVB 5 wt%.

**Figure 11 polymers-16-03472-f011:**
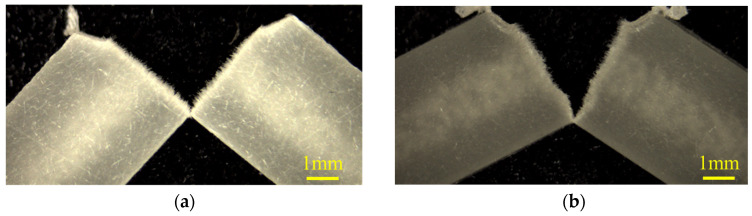
Examples of the specimen following a notched Charpy impact test when GF content is 20 wt%: (**a**) PP/GF and (**b**) PP/GF/MAH-PP.

**Figure 12 polymers-16-03472-f012:**
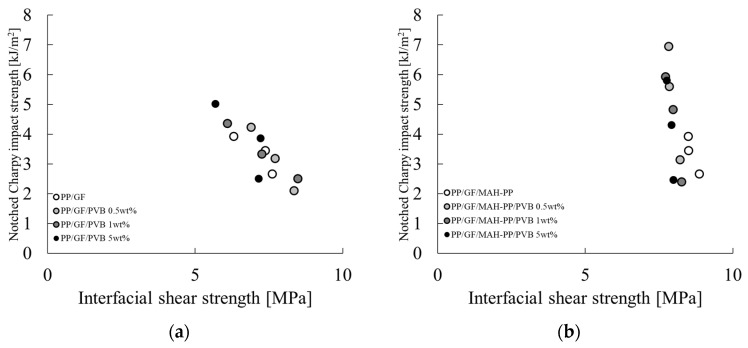
Relations between notched Charpy impact strength and interfacial shear strength of (**a**) PP/GF/PVB and (**b**) PP/GF/MAH-PP/PVB composites.

**Figure 13 polymers-16-03472-f013:**
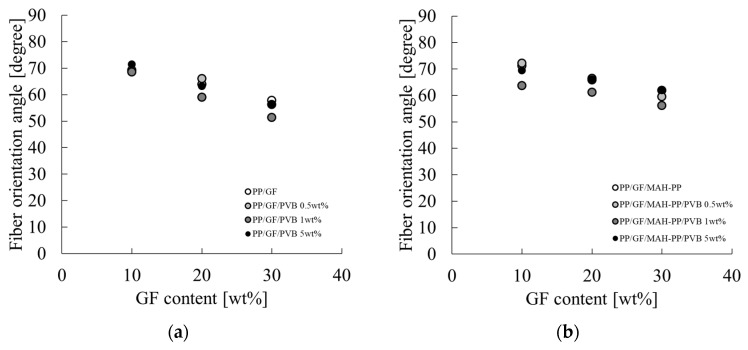
GF content dependences on the averaged fiber orientation angle of (**a**) PP/GF/PVB and (**b**) PP/GF/MAH-PP/PVB composites.

**Figure 14 polymers-16-03472-f014:**
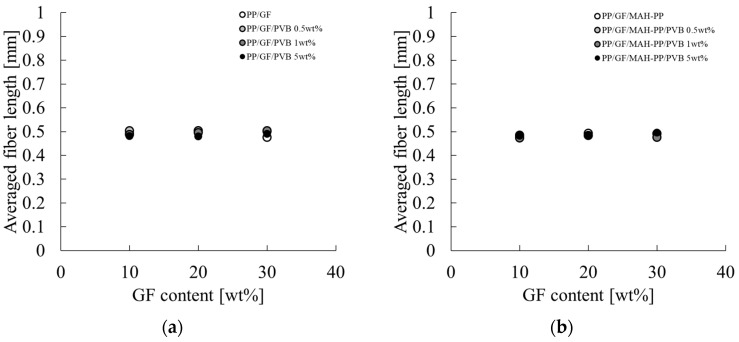
GF content dependences on the averaged fiber length of (**a**) PP/GF/PVB and (**b**) PP/GF/MAH-PP/PVB composites.

**Figure 15 polymers-16-03472-f015:**
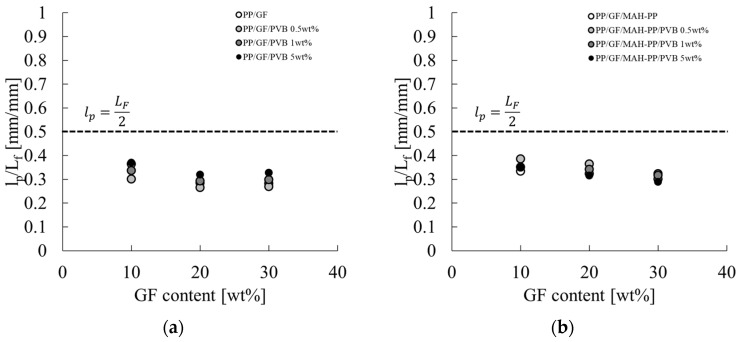
GF content dependences on the l_p_/L_F_ of (**a**) PP/GF/PVB and (**b**) PP/GF/MAH-PP/PVB composites.

**Figure 16 polymers-16-03472-f016:**
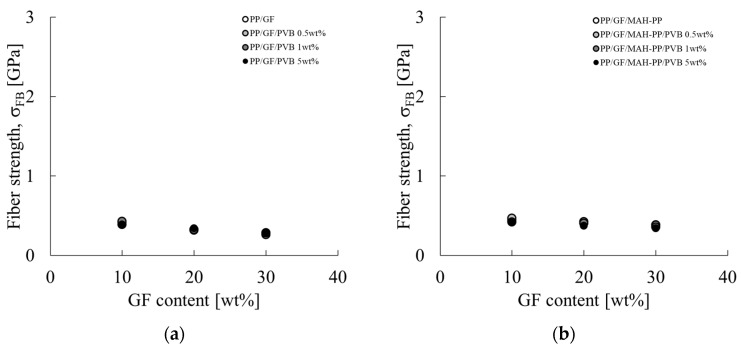
GF content dependences on the fiber strength σ_FB_ of (**a**) PP/GF/PVB and (**b**) PP/GF/MAH-PP/PVB composites obtained from Equation (13).

**Figure 17 polymers-16-03472-f017:**
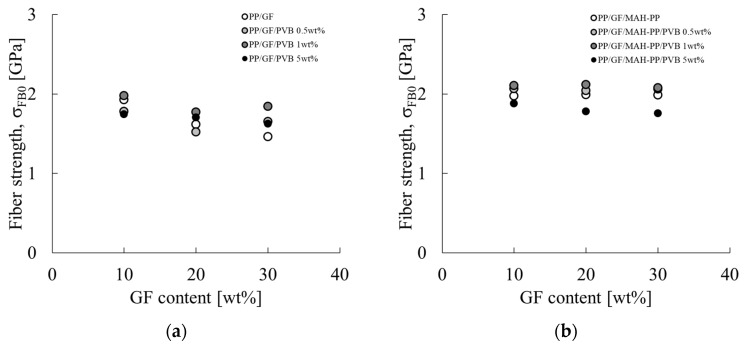
GF content dependences on the fiber strength σ_FB0_ of (**a**) PP/GF/PVB and (**b**) PP/GF/MAH-PP/PVB composites obtained from Equation (16).

**Figure 18 polymers-16-03472-f018:**

Fracture surface observations obtained after notched Charpy impact tests when GF content is 20 wt%. (**a**) PP/GF, (**b**) PP/GF/PVB 0.5 wt%, (**c**) PP/GF/PVB 1 wt%, (**d**) PP/GF/PVB 5 wt%, (**e**) PP/GF/MAH-PP, (**f**) PP/GF/MAH-PP/PVB 0.5 wt%, (**g**) PP/GF/MAH-PP/PVB 1 wt%, and (**h**) PP/GF/MAH-PP/PVB 5 wt%.

**Table 1 polymers-16-03472-t001:** Injection molding condition.

PP (wt%)	GF (wt%)	MAH-PP (wt%)	PVB (wt%)	T_inj _ (°C)	T_mold _ (°C)	V_inj _ (mm/s)	P_hold_ (MPa)	T_inj_ (s)	T_cool_ (s)
90	10	0	0	230	50	10	56	30	15
80	20	0	0	230	50	10	56	30	15
70	30	0	0	230	50	10	56	30	15
89.5	10	0	0.5	230	50	10	52.5	30	15
79.5	20	0	0.5	230	50	10	52.5	30	15
69.5	30	0	0.5	230	50	10	52.5	30	15
89	10	0	1	230	50	10	42	30	15
79	20	0	1	230	50	10	49	30	15
69	30	0	1	230	50	10	49	30	15
85	10	0	5	230	50	10	49	30	15
75	20	0	5	230	50	10	49	30	15
65	30	0	5	230	50	10	56	30	15
85	10	5	0	230	50	10	49	30	15
75	20	5	0	230	50	10	52.5	30	15
65	30	5	0	230	50	10	52.5	30	15
84.5	10	5	0.5	230	50	10	49	30	15
74.5	20	5	0.5	230	50	10	49	30	15
64.5	30	5	0.5	230	50	10	49	30	15
84	10	5	1	230	50	10	42	30	15
74	20	5	1	230	50	10	49	30	15
64	30	5	1	230	50	10	49	30	15
80	10	5	5	230	50	30	49	30	15
70	20	5	5	230	50	30	49	30	15
60	30	5	5	230	50	30	49	30	15

**Table 2 polymers-16-03472-t002:** Characteristic values used for this study.

PP (wt%)	GF (wt%)	MAH-PP (wt%)	PVB (wt%)	IFSS (MPa)	σ_w_ (MPa)	a_iN_ (kJ/m^2^)	γ_i_ (N/mm)	φ (degree)	L_f_ (mm)	σ_FB_ (MPa)	σ_FB0_ (MPa)
90	10	0	0	7.6	22.7	2.7	0.99	68.9	0.49	419	1928
80	20	0	0	7.4	21.4	3.5	0.43	63.9	0.50	326	1618
70	30	0	0	6.3	18.0	3.9	0.20	57.8	0.48	261	1460
89.5	10	0	0.5	8.4	22.4	2.1	0.65	69.2	0.50	389	1782
79.5	20	0	0.5	7.7	20.1	3.2	0.19	66.1	0.50	317	1520
69.5	30	0	0.5	6.9	17.4	4.2	0.04	56.3	0.50	286	1653
89	10	0	1	8.5	23.4	2.5	1.01	68.6	0.49	429	1981
79	20	0	1	7.3	20.2	3.3	0.33	59.0	0.50	325	1772
69	30	0	1	6.1	17.7	4.4	0.14	51.4	0.50	281	1843
85	10	0	5	7.2	20.3	2.5	1.77	71.4	0.48	391	1744
75	20	0	5	7.2	18.4	3.9	0.82	63.2	0.48	340	1706
65	30	0	5	5.7	14.6	5.0	0.48	56.2	0.49	281	1625
85	10	5	0	8.9	25.9	2.6	1.22	71.2	0.49	443	1975
75	20	5	0	8.5	25.3	5.0	0.57	66.5	0.49	418	1990
65	30	5	0	8.5	22.9	6.7	0.31	62.0	0.49	387	1987
84.5	10	5	0.5	8.2	24.8	3.1	0.99	72.2	0.48	468	2066
74.5	20	5	0.5	7.9	24.0	5.6	0.41	65.8	0.48	425	2044
64.5	30	5	0.5	7.8	22.2	6.9	0.20	59.6	0.49	383	2063
84	10	5	1	8.3	26.5	2.4	1.31	63.7	0.47	423	2106
74	20	5	1	8.0	25.6	4.8	0.57	61.1	0.48	406	2119
64	30	5	1	7.7	23.6	5.9	0.30	56.2	0.48	358	2077
80	10	5	5	8.0	25.2	2.5	2.13	69.4	0.48	412	1881
70	20	5	5	7.9	24.4	4.3	1.09	66.0	0.48	372	1780
60	30	5	5	7.8	23.0	5.8	0.71	61.9	0.50	342	1758

**Table 3 polymers-16-03472-t003:** Averaged linear expansion coefficient and density of base materials.

Materials	E_f_	α	Density
	MPa	×10^−4^/K	g/cm^3^
PP	1475	1.01	0.9
PVB	30	22.4	1.10
GF	N.D.	0.05	2.56

## Data Availability

The data presented in this study are sufficient for the purposes of this investigation; however, they are available for further analysis upon request from the corresponding author.

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
