# Peer review of "Polyvinyl Butyral Addition Effects on Notched Charpy Impact Strength of Injection-Molded Glass Fiber-Reinforced Polypropylene"

_polymers, 2024, doi:10.3390/polym16243472_

Round 1

Reviewer 1 Report

Comments and Suggestions for Authors

1. Can include numerical values in the abstract and conclusion and if possible include some latest relevant references. 

2. Technically discuss why there is there is variation in interfacial shear strength based on the GF percent according to figure 1. 

3. Infer the variation in weld strength and impact strength based on the GF percent according to figure 2 and 3. 

4. As shown in the figure 5, interfacial interaction force is high for PP/GF/PVB with wt. percent 5, state the reason. 

5. Justify why there is randomness in the impact strength Vs interfacial shear strength as shown in the figure 7. 

6. GF content dependences on the averaged fiber orientation angle graph reveals least variation, state why? As shown in the figure 8.

7.  Technical discussion by state the reason is missing for figure 9, 10, 11, and 12, please include. 

8. If possible increase the clarity of figure 13.

9. State novelty of the article. 

Comments on the Quality of English Language

The English can be improved for better clarity.

Author Response

Manuscript ID: Polymers-3349543

Title: Polyvinyl Butyral Addition Effects on Notched Charpy impact strength of Injection Molded Glass Fiber Reinforced Polypropylene

Authors: Tetsuo Takayama *, Yuuki Yuasa, Quan Jiang

Thank you for your interest in reviewing this paper. I would like to respond to your suggestions and questions raised below.

  1. Can include numerical values in the abstract and conclusion and if possible include some latest relevant references. 

Thank you for bringing this to our attention. We have revised the abstract and conclusion to address this issue. With regard to the literature, we have included the results of our research studies in the References section, to the best of our ability at this stage.

  1. Technically discuss why there is there is variation in interfacial shear strength based on the GF percent according to figure 1. 

Thank you for bringing this to our attention. We have added an explanation of the changes in IFSS with increasing GF content to the relevant section of the document.

  1. Infer the variation in weld strength and impact strength based on the GF percent according to figure 2 and 3. 

Thank you for bringing this to our attention. We have added explanations for the changes in weld strength and impact strength with increasing GF content, respectively, to address this issue.

  1. As shown in the figure 5, interfacial interaction force is high for PP/GF/PVB with wt. percent 5, state the reason. 

Thank you for bringing this to our attention. We have added an explanation of the change in the interfacial interaction force when 5 wt% PVB is added to the document.

  1. Justify why there is randomness in the impact strength Vs interfacial shear strength as shown in the figure 7. 

Thank you for bringing this to our attention. We have added a discussion of the variation seen in Figure 7 to the document.

  1. GF content dependences on the averaged fiber orientation angle graph reveals least variation, state why? As shown in the figure 8.

Thank you for your thoughtful suggestion. We have added a discussion on the fiber orientation angle trend observed in Figure 8, which we hope will be of interest to you.

  1. Technical discussion by state the reason is missing for figure 9, 10, 11, and 12, please include. 

I'm grateful to you for bringing this to my attention. Regarding the figures you kindly highlighted, we believe that Figures 10 and 11, which show the results obtained during the analysis of this paper, speak for themselves. We hope that you will find the additional discussion of these results unnecessary.

Regarding Figures 9 and 12, we have taken the liberty of adding explanations and discussions.

  1. If possible increase the clarity of figure 13.

I'm grateful to you for bringing this to my attention. It is challenging to make adjustments to the SEM photographs due to the limitations of the equipment's resolution.

  1. State novelty of the article.

This paper introduces a novel approach by utilizing PVB extracted from waste automobiles as an additive. As a result, the authors were able to achieve a sizing effect, in which glass fibers are dispersed in a bundle, and highlighted that this may contribute to the improvement of notched impact strength. We believe that this is one of the few results that can potentially enhance the properties of existing materials by effectively utilizing waste materials. It is our hope that the publication of these results in this journal as an academic paper will serve as a valuable reference for future material development research.

We are grateful to have had the opportunity to have our paper reviewed by additional native speakers, and we are pleased to report that their feedback has been very helpful.

Reviewer 2 Report

Comments and Suggestions for Authors

In this study, the mechanical properties of GSFRTP with a PP matrix were evaluated when PVB was added. The mechanical properties of GSFRTP were comprehensively studied, and the mechanism of improvement via PVB was revealed. The results demonstrated that the combination of PVB and maleic anhydride-modified PP with GSFRTP can increase the notched impact strength. The study may promote the sustainability of PVB and the application of GSFRTP. Furthermore, this paper was written with explicit language, and good organization. Thus, I suggest minor revisions according to the following comments.

1.     Add some figures of the details of experiment, such as specimen, test setup, etc.

2.     Provide some pictures describing injection moulding.

3.     Provide the details of fibers and resin, such as strength, elastic modulus, etc.

4.     Add a table to show the details of test groups.

5.     Provide pictures showing the failure modes of specimens.

6.     Some important quantitative data of the study are suggested to be added in the Abstract.

7.     The Summary is too short. Generally, the main conclusions of the article should be stated in detail and quantitatively.

8.     A literature is recommended as reference:

Fiber-reinforced polymers and fiber-reinforced concrete in civil engineering. Buildings, 2023,13(7): 1755

Author Response

Manuscript ID: Polymers-3349543

Title: Polyvinyl Butyral Addition Effects on Notched Charpy impact strength
of Injection Molded Glass Fiber Reinforced Polypropylene

Authors: Tetsuo Takayama *, Yuuki Yuasa, Quan Jiang

Thank you for your interest in reviewing this paper. I would like to respond to your suggestions and questions raised below.

In this study, the mechanical properties of GSFRTP with a PP matrix were evaluated when PVB was added. The mechanical properties of GSFRTP were comprehensively studied, and the mechanism of improvement via PVB was revealed. The results demonstrated that the combination of PVB and maleic anhydride-modified PP with GSFRTP can increase the notched impact strength. The study may promote the sustainability of PVB and the application of GSFRTP. Furthermore, this paper was written with explicit language, and good organization. Thus, I suggest minor revisions according to the following comments.

  1. Add some figures of the details of experiment, such as specimen, test setup, etc.

Your observation is much appreciated. In order to provide a more comprehensive picture, additional figures have been included for short beam shear tests, tensile tests, and notched Charpy impact tests.

  1. Provide some pictures describing injection moulding.

I'm grateful to you for bringing this to my attention. We have taken the liberty of adding a photo of the exterior of the injection molding machine used.

  1. Provide the details of fibers and resin, such as strength, elastic modulus, etc.

Thank you for bringing this to our attention. Table 3 provides details on the properties of the PP, PVB, and GF used. We have added information on the melt volume rate of PP, MAH-PP, and PVB used, as well as the fiber diameter and length of GF before compounding.

  1. Add a table to show the details of test groups.

I'm grateful to you for bringing this to my attention. We have taken the liberty of including the numerical data used in the figures in this paper as Table 2 for your convenience.

  1. Provide pictures showing the failure modes of specimens.

Thank you for the information. I have added an overview photo of the specimen after the Charpy impact test as a figure, as suggested.

  1. Some important quantitative data of the study are suggested to be added in the Abstract.

I'm grateful to you for bringing this to my attention. We have taken the liberty of revising the abstract.

  1. The Summary is too short. Generally, the main conclusions of the article should be stated in detail and quantitatively.

I appreciate you bringing this to my attention. I have added a description of the summary with specific numbers to address this issue.

  1. A literature is recommended as reference:

Fiber-reinforced polymers and fiber-reinforced concrete in civil engineering. Buildings, 2023,13(7): 1755

I'm grateful to you for bringing this to my attention. We have taken the liberty of reading about the paper you so kindly recommended. As the content was not sufficiently detailed to be included in this paper, we kindly ask that you refrain from citing it at this time.

We are grateful to have had the opportunity to have our paper reviewed by additional native speakers, and we are pleased to report that their feedback has been very helpful.

Reviewer 3 Report

Comments and Suggestions for Authors

After carefully check the manuscript quality, read the results and how the results were described and presented, especially in the form of SEM image showed the fiber failure, I recommend the present version can be accepted.

Author Response

Thank you for reviewing this paper.

We appreciate your acceptance of the content of this paper.

Reviewer 4 Report

Comments and Suggestions for Authors

Review Report

Manuscript: Polyvinyl Butyral Addition Effects on Notched Charpy impact strength of Injection Molded Glass Fiber Reinforced Polypropylene

Journal: Polymers- MDPI

While the authors have prepared a commendable manuscript, several modifications are needed to ensure its suitability for publishing in "Journal of Polymers- MDPI."

The below notices outline the necessary revisions for the improved version:

1.     Page 1, lines 37-40: The sentence is longer than usual, and the information in it is not supported by any reference.

2.     Page 1, lines 61-63: The authors stated that when the GSFRTP is melt-molded again, the mechanical properties will be reduced. However, they did not give an explanation for this decrease in mechanical properties, nor did they specify what these properties are.

3.     Although the Charpy impact test is included in the title of the manuscript, it is missing from the abstract, introduction, and literature review. There should have been a paragraph explaining the importance of the Charpy impact test for studies of Glass short fiber-reinforced thermoplastics (GSFRTP).

4.     The study included many tests, why was the focus on Charpy impact tests rather than the other tests in the title.

5.     Some equations included in the manuscript are not cited.

6.     In academic articles, the term "numerical data" refers to data obtained by various methods such as finite difference and finite element methods. However, neither these methods nor the software package used are mentioned in this manuscript.

7.     The "weld" term is usually used to refer to fusion welding or solid-state welding. Normally the joining of polymers and composite materials referred by "joining". Could you provide some valuable references that use the term "weld" in the connection of fiber-reinforced thermoplastics?

8.     The last part of the manuscript, as is customary in academic articles, is called (conclusions) instead of (summary), and is often an introductory paragraph followed by the most important conclusions in the form of sequential points.

9.     It is recommended to avoid referencing Figure 2 multiple times within a short sentence. Consider rephrasing the sentence for clarity and conciseness, such as: "The specimens were positioned according to the orientation shown in the schematic representation of the test (Figure 2)."

10.  Do not refer to more than three references in one sentence without mentioning essential differences, as happened in [17–21].

11.  There are a set of references starting from 27 in which there are differences between the text of the manuscript and the list of references.

Author Response

Thank you for your interest in reviewing this paper. I would like to respond to your suggestions and questions raised below.

  1. Page 1, lines 37-40: The sentence is longer than usual, and the information in it is not supported by any reference.

> I appreciate your bringing this to my attention. I have divided this text into two sections and included the review article on GSFRTP as a reference.

  1. Page 1, lines 61-63: The authors stated that when the GSFRTP is melt-molded again, the mechanical properties will be reduced. However, they did not give an explanation for this decrease in mechanical properties, nor did they specify what these properties are.

> Thank you for bringing this to our attention. We have replaced the term "mechanical properties" with a description of specific characteristic values to avoid any potential ambiguity.

  1. Although the Charpy impact test is included in the title of the manuscript, it is missing from the abstract, introduction, and literature review. There should have been a paragraph explaining the importance of the Charpy impact test for studies of Glass short fiber-reinforced thermoplastics (GSFRTP).

> Your observation is much appreciated. In light of the fact that notched impact strength plays a significant role in determining fiber length, we have taken the liberty of including that information in the abstract and introduction.

  1. The study included many tests, why was the focus on Charpy impact tests rather than the other tests in the title.

> Your observation is much appreciated. In this paper, we concentrated on notched Charpy impact strength as it is the most crucial property of the GSFRTP product. To analyse this property, we conducted experiments to obtain information on interfacial shear strength, fiber length, and fiber orientation angle.

  1. Some equations included in the manuscript are not cited.

> Thank you for bringing this to our attention. We have added bibliography numbers to all the formulas taken from the literature. The formulas that are not numbered are those proposed in this paper.

  1. In academic articles, the term "numerical data" refers to data obtained by various methods such as finite difference and finite element methods. However, neither these methods nor the software package used are mentioned in this manuscript.

> Thank you for bringing this to our attention. We recognize that "numerical data" may not be the most appropriate term in this context. We will replace it with "characteristic values" to better align with the intended meaning.

  1. The "weld" term is usually used to refer to fusion welding or solid-state welding. Normally the joining of polymers and composite materials referred by "joining". Could you provide some valuable references that use the term "weld" in the connection of fiber-reinforced thermoplastics?

> I appreciate your bringing this to my attention. One of the molding defects that can occur during injection molding is weld lines. These are formed when molten resin merges during injection molding. The joining phenomenon that occurs at this time is similar to fusion welding, which is why the term "weld" is used. I believe that the term "joining" is used to describe the joining of two separate solid molded products into one, which is not what I wanted to express in this paper. The wording has been changed to clear up misunderstandings.

  1. The last part of the manuscript, as is customary in academic articles, is called (conclusions) instead of (summary), and is often an introductory paragraph followed by the most important conclusions in the form of sequential points.

> Thank you for pointing this out, I have changed it to Conclusions.

  1. It is recommended to avoid referencing Figure 2 multiple times within a short sentence. Consider rephrasing the sentence for clarity and conciseness, such as: "The specimens were positioned according to the orientation shown in the schematic representation of the test (Figure 2)."

> Thank you for pointing this out. We have corrected it as you indicated.

  1. Do not refer to more than three references in one sentence without mentioning essential differences, as happened in [17–21].

> Thank you for pointing this out. We have reviewed the literature and made corrections to the text to follow your suggestion.

  1. There are a set of references starting from 27 in which there are differences between the text of the manuscript and the list of references.

> Thank you for pointing this out. We have corrected it to the correct literature number.

Round 2

Reviewer 4 Report

Comments and Suggestions for Authors

The concerns raised in the previous review of manuscript "Polyvinyl Butyral Addition Effects on Notched Charpy impact strength of Injection Molded Glass Fiber Reinforced Polypropylene" have been successfully addressed by the authors.

Additionally, certain scientific terms, such as numerical values and welding-related terminology, have been corrected to align with academic writing standards.

Therefore, I consider this manuscript suitable for publication in its current form in the Polymers journal.